# Mammalian Homologue NME3 of DYNAMO1 Regulates Peroxisome Division

**DOI:** 10.3390/ijms21218040

**Published:** 2020-10-28

**Authors:** Masanori Honsho, Yuichi Abe, Yuuta Imoto, Zee-Fen Chang, Hanna Mandel, Tzipora C. Falik-Zaccai, Yukio Fujiki

**Affiliations:** 1Medical Institute of Bioregulation, Institute of Rheological Functions of Food—Kyushu University Collaboration Program, Kyushu University, 3-1-1 Maidashi, Fukuoka 812-8582, Japan; mhonsho@med.kyushu-u.ac.jp (M.H.); abe.yuichi.439@m.kyushu-u.ac.jp (Y.A.); 2Faculty of Arts and Science, Kyushu University, 744 Motooka, Fukuoka 819-0395, Japan; 3Department of Cell Biology, School of Medicine, Johns Hopkins University, Baltimore, MD 21205, USA; yimoto1@jhu.edu; 4Institute of Molecular Medicine, College of Medicine, National Taiwan University, Taipei 10002, Taiwan; zfchang@ntu.edu.tw; 5Galilee Medical Center, Institute of Human Genetics, Nahariya 22100, Israel; h_mandel@rambam.health.gov.il (H.M.); falikmd.genetics@gmail.com (T.C.F.-Z.); 6Azrieli Faculty of Medicine, Bar-Ilan University, Safed 13100, Israel

**Keywords:** peroxisome, constriction, NDP kinase, GTP, *nme3* patient

## Abstract

Peroxisomes proliferate by sequential processes comprising elongation, constriction, and scission of peroxisomal membrane. It is known that the constriction step is mediated by a GTPase named dynamin-like protein 1 (DLP1) upon efficient loading of GTP. However, mechanism of fuelling GTP to DLP1 remains unknown in mammals. We earlier show that nucleoside diphosphate (NDP) kinase-like protein, termed dynamin-based ring motive-force organizer 1 (DYNAMO1), generates GTP for DLP1 in a red alga, *Cyanidioschyzon merolae.* In the present study, we identified that nucleoside diphosphate kinase 3 (NME3), a mammalian homologue of DYNAMO1, localizes to peroxisomes. Elongated peroxisomes were observed in cells with suppressed expression of *NME3* and fibroblasts from a patient lacking NME3 due to the homozygous mutation at the initiation codon of *NME3*. Peroxisomes proliferated by elevation of NME3 upon silencing the expression of ATPase family AAA domain containing 1, *ATAD1*. In the wild-type cells expressing catalytically-inactive NME3, peroxisomes were elongated. These results suggest that NME3 plays an important role in peroxisome division in a manner dependent on its NDP kinase activity. Moreover, the impairment of peroxisome division reduces the level of ether-linked glycerophospholipids, ethanolamine plasmalogens, implying the physiological importance of regulation of peroxisome morphology.

## 1. Introduction

Peroxisomes proliferate by division involving elongation, constriction, and fission [1,2,3]. Peroxisome division is mediated by several factors including Pex11β, dynamin-like protein 1 (DLP1) [4,5], mitochondrial fission factor (Mff) [6,7], and Fission 1 (Fis1) [8,9] in mammals [10]. Except for Pex11β, these proteins are originally identified as fission factors of mitochondria, indicating that peroxisomes share a common division-machinery with mitochondria [2,8,11]. Knocking out of *PEX11β* in mouse [12] and genetic defect of human *PEX11β* decrease peroxisome abundance [13,14]. N-terminal amphipathic helix region of Pex11β is required for the formation of Pex11β homo-oligomer and interaction with membrane phospholipids, leading to deformation of peroxisomal membrane [15,16,17]. Docosahexaenoic acid, a poly-unsaturated fatty acid of peroxisomal β-oxidation metabolites, induces elongation of peroxisomes, hyper-oligomerization of Pex11β on the elongated regions, and extension of Pex11β-enriched membrane [7].

DLP1, a member of the dynamin GTPase family, is essential for membrane fission of peroxisomes and mitochondria by functioning at the membrane constriction sites [5,18,19,20,21,22]. Pex11β forms a ternary fission machinery complex with Mff and DLP1 at the constricted membrane region of the elongated peroxisomes, promoting fission during the peroxisome division [3,23]. Pex11β enhances GTPase activity of DLP1 [24], indicating the multiple roles of Pex11β throughout processes of peroxisome division. DLP1 has a low affinity to and a high hydrolysis rate of GTP [25], suggesting the local GTP-loading is important for the function of DLP1. However, mechanisms underlying the GTP loading to DLP1 remain unknown in mammals.

During the division of peroxisomes, DLP1 polymerizes and forms a ring or spiral structure, called peroxisome-dividing machinery, to constrict and pinch off the peroxisomal membrane [26]. By proteomic analysis of highly purified peroxisome-dividing machinery of a unicellular red alga, *Cyanidioschyzon merolae*, we recently identified 17-kDa nucleoside diphosphate (NDP) kinase-like protein, termed dynamin-based ring motive-force organizer 1 (DYNAMO1) [27]. DYNAMO1 colocalizes with Dnm1, a *C. merolae* homologue of human DLP1, and is identified as an essential component of mitochondria- and peroxisome-dividing machineries, where it locally generates GTP from ATP and GDP for DLP1 [27]. In ten mammalian NDP kinases so far identified, nucleoside diphosphate kinase 3 (NME3) encoded by *NME3* gene belonging to the group I NDP kinase may be a potential candidate for a mammalian orthologue of DYNAMO1, based on the similarity of amino-acid sequences and localization of NME3 protein at outer membrane of mitochondria when fused with GFP [28]. A patient with NME3 deficiency shows a phenotype that a reduced level of mitochondrial fusion, resulting in a fatal neurodegenerative disorder [29]. Here, we report that NME3 is indeed localized to peroxisomes and participates in peroxisome morphogenesis in human cells.

## 2. Results

### 2.1. Morphological Changes of Peroxisomes in Cells Reduced Expression of NME3

In *C. merolae*, DYNAMO1 fuels DLP1 with GTP by locally generating GTP from ATP and GDP [27]. In the mammalian NME protein family, NME1 and NME2 localize to clathrin-coated pits (CCPs) and fuel GTP to dynamin (Figure 1) [28]. NME4 is shown to interact with OpaI, whose GTPase domain is located in the intermembrane space of mitochondria (Figure 1) [30,31,32]. Given the localization of NME3 in mitochondrial outer membrane [28,29] and a high degree of similarity of the primary sequences between DYNAMO1 and NME3 (Figure 1), we suspected that NME3 localizes to peroxisomes as well as mitochondria to regulate their fission together with DLP1 and Mff, both mediate the fission of peroxisomes and mitochondria [4,5,6,7]. Therefore, we investigated the morphology of peroxisomes in F741 fibroblasts from a patient carrying the homozygous mutation in the initiation codon of *NME3* (Figure 2). In fibroblasts from the patient F741, import of peroxisomal matrix protein was not affected as judged by the localization of peroxisomal matrix proteins including catalase, PTS1-proteins, and a PTS2-protein, alkyldihydroxyacetonephosphate synthase (ADAPS) (Figure 2A–C). Consistent with these results, proteolytic processing of acyl-CoA oxidase (AOx), ADAPS and another PTS2-protein, 3-ketoacyl-CoA thiolase (thiolase) [33,34], mediated by trypsin domain-containing 1 [35,36] was discernible as observed in fibroblasts from a healthy control (Figure 2D), although the protein level of AOx-A chain and thiolase was relatively higher than those in control fibroblasts. These results suggest that the import of peroxisomal matrix proteins is not affected in the absence of NME3. Interestingly, slightly elongated peroxisomes were frequently observed in the patient’s fibroblasts by immunofluorescence microscopic analysis using antibodies to peroxisomal matrix proteins including catalase as well as Pex14 (Figure 2A–C), which is ascertained by measuring the length of each peroxisome (Figure 2E), although the elongated peroxisomes are reported to be more evident in fibroblasts from patients with a homozygous mutation in the genes encoding DLP1 and Mff [21,37,38]. The elongated peroxisomes were also observed in HeLa cells transfected with three different dsRNAs against *NME3* (Figure 3A) as quantified the elongated peroxisomes (Figure 3C), where the level of *NME3* mRNA were significantly reduced as compared with those of untreated HeLa cells (Figure 3B). Taken together, these results suggest that NME3 is involved in the morphogenesis of peroxisomes as well.

### 2.2. Intracellular Localization of NME3

From the finding that a reduced level of expression and the absence of NME3 cause the elongation of peroxisomes (Figure 2 and Figure 3), we suspected that NME3 localizes to peroxisomes and mediates their fission. Mitochondrial localization of exogenously expressed NME3-HA_2_ was observed in HeLa cells (Figure 4A, lower row), as shown by expressing NME3-GFP or FLAG-tagged NME3 (FLAG-NME3) [28,29]. In addition, NME3-HA_2_-positive punctate structures were stained with Pex14, indicative of peroxisomes, where mitochondria were absent as judged by the signal of Tom20 (Figure 4A, upper row). However, NME3-HA_2_ signal was not discernible in most of peroxisomes (Figure 4A, upper row), which might be due to a lower targeting-efficiency of NME3-HA_2_ to peroxisomes and/or a high turnover of NME3-HA_2_ on peroxisomes despite of the detectable expression of NME3-HA_2_ as judged by immunoblotting with anti-NME3 antibody against the cytosolic catalytic domain of NME3 (Figure 4B). Endogenous NME3 in HeLa cells was not readily detectable with this anti-NME3 antibody, apparently due to a very low level of NME3 expression [29] (Figure 4B, lane 4).

Next, we assessed the complex formation of NME3-HA_2_ with the cytosolic membrane protein receptor, Pex19, in the cytosol, which is well-characterized step during the transport of peroxisomal membrane proteins prior to targeting to peroxisomes [40,41,42]. We followed the previously established method to monitor the amount of peroxisomal membrane protein in the cytosol by the co-expression of Pex19 [41,42]. Upon expressing HA-tagged Pex26, a peroxisomal C-tail anchored protein, with FLAG-Pex19, Pex26-HA_2_ was recovered more in the cytosolic fractions than that expressed alone, suggesting that the complexes of FLAG-Pex19 with Pex26-HA_2_ were formed in the cytosol (Figure 5A,C). Similarly, amount of NME3-HA_2_ in the cytosolic fraction was elevated by co-expression with FLAG-Pex19 (Figure 5B,D). Together, these results suggest that NME3-HA_2_ forms a complex with Pex19 in the cytosol and targeted/transported to peroxisomes via the Pex19-Pex3 pathway.

We further assessed the intracellular localization of NME3 by expressing non-tagged NME3 in HeLa cells. As anticipated, mitochondrial localization of NME3 was discernible by the antibody raised to DYNAMO1, presumably due to a high degree of the similarity in the primary sequences between DYNAMO1 and NME3 (see Figure 1B), where the tubular network of mitochondria was not discernible (Figure 6). In the same cells, punctate immunofluorescence signals for NME3 merged with Pex14 but not Tom20 were discernible (Figure 6).

### 2.3. NME3 Is Elevated by Knockdown of ATAD1

NME3 and NME3-HA_2_ were partly localized to peroxisomes (Figure 4A and Figure 6), whereas fission proteins such as Mff and Fis1 are widely localized to peroxisomes [8,9,11]. Based on these results, we suspected that C-terminal tagging suppresses peroxisomal targeting of NME3-HA_2_ and/or the protein level of NME3 is post-translationally regulated on peroxisomes. We examined a possibility whether ATPase family AAA domain-containing protein 1 (ATAD1) regulates the turnover of NME3, because ATAD1 is known to play a role in the elimination of membrane protein mitochondrially mislocalized C-tail anchored proteins such as GOS28, peroxisomal Pex26, and Pex15 [43], maybe including other peroxisomal membrane proteins. To our surprise, the protein level, but not at the transcription level, of endogenous NME3 was elevated and peroxisomal localization of NME3 became readily discernible by knocking down *ATAD1*, but not in control HeLa cells (Figure 7A,C,E–G), where the protein level of Pex14 was not altered (Figure 7H), implying that ATAD1 is involved in regulating the expression level of NME3. The elevation of NME3 by ATAD1 knockdown was confirmed by ectopic expression of NME3 in HeLa cells, where about 60% decrease in *ATAD1* mRNA and ~1.4-fold increase in NME3-1 were detectable (Figure 7J), consistent with the profile of endogenous NME3 (Figure 7E–G,I). The immunofluorescence signal of NME3 merged with Tom20 was discernible in HeLa cells with anti-DYNAMO1 antibody, indicative of mitochondrial localization, which was then diminished by the knockdown of *NME3* (Figure 7A,B). Peroxisomal localization of NME3 was observed in HeLa cells that had been treated for knocking down *ATAD1*, but not control HeLa cells (Figure 7A,C). Taken together, these results suggest that NME3 is stably localized to peroxisomes in ATAD1-suppressed cells.

### 2.4. Peroxisomes Are Increased in Number upon Knocking down ATAD1

Peroxisomes are slightly elongated in F741 fibroblasts and HeLa cells suppressed in the expression of *NME3* (Figure 2 and Figure 3). Furthermore, in *PEX11β*-knocked out mouse embryonic fibroblasts (MEF) [12], ectopically expressed NME3 is observed at an apparent constriction site where DLP1 was accumulated but signal of Pex14 is weak, and the both sides adjacent to the constriction site in an elongated peroxisome (Figure 7K). These results resemble the Pex11β-enriched constriction site formed along the elongated peroxisome devoid of Pex14 [7], suggesting that NME3 is involved in a fission step of peroxisomes. Noteworthily, the elongated peroxisomes are more evident in fibroblasts from the patients defective in *DLP1* and *Mff* [21,37,38]. Interestingly, overexpression of Pex11β but not DLP1 or Mff promotes peroxisome division, leading to an increase in the number of peroxisomes [4,11,44]. Given these findings and together with the enhanced expression of NME3 by the *ATAD1* knockdown (Figure 7), we analyzed the abundance of peroxisomes in HeLa cells. In *ATAD1*-knocked down HeLa cells, the number of peroxisomes was increased by ~40% as compared to that in mock-treated HeLa cells, which was returned to the control level by co-suppression of *NME3* (Figure 7D,I), where peroxisomes were often elongated (Figure 7D). These results suggest that the elevated protein level of NME3 in peroxisomes is responsible for the increase in the number of peroxisomes in *ATAD1*-knocked down HeLa cells.

### 2.5. Peroxisomes Are Elongated by Expression of Catalytically Inactive NME3

Elevation of NME3 protein level facilitated peroxisome division (Figure 7). We next examined whether NDP kinase activity of NME3 is required for the division of peroxisomes. To this end, we expressed a catalytically inactive mutant NME3, termed NME3H135N, where the catalytic site histidine at amino-acid position 135 had been mutated to asparagine (NME3H135N). Mutation of the catalytic site histidine in NME3 and DYNAMO1 abolishes their NDP kinase activity [27,29]. In wild-type NME3-expressing HeLa cells, double protein-bands were detected with anti-NME3C antibody raised against the C-terminal 18 amino-acid sequence of NME3 (Figure 8A, lane 2). However, upon tagging of a tandem HA peptide at the C-terminus of NME3, the NME3-HA_2_ protein was no longer detectable with the NME3C antibody, implying that the NME3C antibody specifically recognizes the amino-acid sequence harboring the free carboxyl group of the peptide derived from NME3 (Figure 8A, lane 3).

The expressed level of NME3H135N was about a half of the wild-type NME3, where the NME3H135N band appeared to be with the same migration as the band with “a lower mobility” of NME3-1 (Figure 8A, lane 4). Under this condition, peroxisomes were frequently elongated in HeLa cells expressing NME3H135N (Figure 8B). We interpret this result to mean that NME3H135N affected division of peroxisomes by interfering with the endogenous NME3, hence suggesting that NDP kinase activity of NME3 is required for the division of peroxisomes.

We further characterized NME3 by addressing the difference between the two forms of NME3, NME3-1 and NME3-2. We investigated intracellular distribution of NME3 by subcellular fractionation assay (Figure 8C). Tom20 was mostly detected in a heavy mitochondria (HM) fraction, whereas Pex3 was mainly in a post-HM membrane fraction (Figure 8C,D). In contrast to the differential distribution of mitochondrial and peroxisomal marker proteins, Tom20 and Pex3, respectively, NME3-1 was recovered in both HM fraction and post-HM fraction containing light mitochondria (LM) and microsome (Ms) fractions, and a small amount of NME3-1 was detected in the cytosol fraction (Figure 8C,D). Cell-free synthesized NME3 migrated at the same position as the NME3-1 band (Figure 8F). However, NME3-2 with a higher mobility was mostly recovered in the post-HM membrane fraction, similar to Pex3 (Figure 8C,D). Contrary to this, NME3H135N was recovered mostly in HM and post-HM fractions, similar to NME3-1, and partly in the cytosol fraction (Appendix A). Interestingly, two additional NME3H135N bands at 20 kDa, termed NME3H135N-a and NME3H135N-b, respectively, were detected in PNS fraction (Appendix A). Both proteins were mainly recovered in a post-HM fraction, suggesting that NME3H135N is likely post-translationally modified on peroxisomes (Appendix A). NME3H135N level was increased like NME3 by ATAD1 knockdown (Appendix A), where NME3H135N and NME3H135N-b were detected. Furthermore, both of NME3-1 and NME3-2 were largely resistant to the sodium carbonate extraction, similar to Pex3, an integral membrane protein residing on peroxisomes (Figure 8E) [45]. NME3H135N was likewise recovered in the membrane fraction upon the alkaline extraction, while both NME3H135N-a and NME3H135N-b were recovered in both membrane and soluble fractions (Appendix A), postulating a less membrane association of the modified forms of NME3H135N. Together, these results suggest that NME3 is localized to both mitochondria and peroxisomes as an integral membrane protein, where the NME3-2 seems to be post-translationally modified on peroxisomes in a manner dependent on histidine at amino-acid position 135, thereby detectable with an apparently higher mass in SDS-PAGE.

### 2.6. Decreased Level of Plasmalogens in F741 Patient Fibroblasts

Biosynthesis of plasmalogens is initiated in peroxisomes [46]. The level of ethanolamine plasmalogens (PlsEtn) is severely reduced in peroxisome-deficient fibroblasts from patients with Zellweger spectrum disorders [47,48]. To investigate whether the milder morphological alteration of peroxisomes in F741 fibroblasts affects peroxisomal lipid metabolism, PlsEtn level in control and F741 fibroblasts were determined by liquid chromatography connected to tandem mass spectrometry (LC-MS/MS). Total amount of PlsEtn in F741 fibroblasts was reduced to about 50% of that in fibroblasts from a healthy control (Figure 9), suggesting that biosynthesis of PlsEtn was attenuated by the morphological alternation of peroxisomes.

## 3. Discussion

In the present study, we show that NME3 localizes to peroxisomes as well as mitochondria. Silencing of *NME3* expression in HeLa cells causes the elongation of peroxisomes, which is also observed in the fibroblasts from NME3-deficient patient devoid of the expression of NME3. An elevated protein level of NME3 by the knockdown of *ATAD1* in HeLa cells leads to an increase in peroxisome number. Moreover, synthesis of plasmalogens was affected in the patient’s fibroblasts. These results indicate that NME3 is involved in the morphogenesis and functions of peroxisomes.

The findings of the localization of NME3 to peroxisomes and mitochondria (Figure 4, Figure 6 and Figure 7) and the NDP kinase activity of recombinant NME3 [29] suggest that NME3 provides GTP to the GTP-requiring proteins localized in peroxisomes and mitochondria, as in the case of DYNAMO1 in *C. merolae* [27]. In mitochondria, NME3 interacts with mitofusin (MFN), a GTPase essential for mitochondria fusion [29]. However, the catalytically inactive NME3 appears to restore the impaired MFN-dependent mitochondrial elongation, suggesting that other GTPases are potential acceptor(s) for GTP provided by NME3. Our morphological results showing the elongated peroxisomes in HeLa cells silenced in the *NME3* expression and the *nme3* patient’s fibroblasts suggest that NME3 more likely provides GTP to GTPases required for fission of peroxisomes. Interestingly, the elongated peroxisomes are formed by the expression of catalytically inactive NME3 and the elevation of NME3 expression level increases the number of peroxisomes, similar to the peroxisomal phenotype reported in the earlier studies by exogenous expression of Pex11β in wild-type cells [9,11,44,49]. Noteworthily, Pex11β functions in the formation of constriction sites in a manner dependent on DHA-containing glycerophospholipids [7] and plays as a GTPase-activating protein of DLP1 [24]. Therefore, NME3 most likely provides GTP to DLP1 for the fission of peroxisomes in mammals as well, as for the DYNAMO1 in *C. merolae* [27], although the interaction between NME3 and DLP1 is not observed by in vivo proximity ligation assay [29]. This is partly due to the limitation of antibodies available for the analysis of in vivo proximity ligation assay, where anti-DLP1 antibody recognizes DLP1 that shuttles between the cytosol and peroxisomes or mitochondrial outer membranes, while NME3 was detected with anti-FLAG antibody recognizing the N-terminus of FLAG-NME3 presumably located in the intermembrane space of mitochondria. We observed the elongated peroxisomes in fibroblasts and HeLa cells devoid of NME3, although the elongated peroxisomes are more evident in the fibroblasts from patients with a homozygous mutation in the genes encoding DLP1 or Mff [21,37,38]. Depletion of both NME1 and NME2 causes the accumulation of invaginated clathrin-coated pits (CCPs) with about 200 nm in length [28], while 300–400 nm-long CCPs are accumulated in dynamin-knockout cells [50,51]. Given these observations, it is conceivable that less tubular phenotype of peroxisomes and CCPs by the NME3-knockdown or -knockout cells are partly due to the utilization of cytosolic GTP. Division of peroxisomes is spatially and temporally well-regulated, hence more detailed analyses would be required for addressing the interaction of NME3 with DLP1 and the effect of NME3 on the GTPase activity of DLP1, leading to the delineation of the molecular mechanisms underlying the division of peroxisomes.

Fission of peroxisome and mitochondrion in *C. merolae* is highly dependent on DYNAMO1 that locates in the cytosol at G1 phase and sequentially localizes to the division sites of mitochondrion and peroxisome [27]. In mammals, NME3 is localized to peroxisomes and mitochondria as an integral membrane protein (Figure 8E). Therefore, the recruitment of NME and DYNAMO1 at the division sites is regulated, likely in a different manner. The protein level of NME3 was elevated by the knockdown of *ATAD1* (Figure 7). The role of ATAD1 in degradation of the mislocalized C-tail anchored proteins has recently been well defined [43,52,53]. The N-terminal hydrophobic segment of NME3 and digestion of the catalytic domain of NME3 by exogenously added proteinase [29] suggest that NME3 is a type I integral membrane protein localizing to peroxisomes and mitochondrial outer membrane. NME3 appears to be a potential clue, together with ATAD1, to investigate molecular mechanisms underlying the regulation in division of peroxisomes and mitochondria.

LC-MS/MS analysis revealed that the level of PlsEtn was decreased in the F741 fibroblasts (Figure 9). Impairments of PlsEtn metabolism have also been reported in other cells with abnormal peroxisome morphology. In *Pex11β^−/−^* mouse with elongated peroxisomes, PlsEtn level in brain is reduced to about 80% of that in control level [12]. *Dlp1*-deficient ZP121 CHO mutant showing tubular peroxisomes and dysmorphology of mitochondria shows the impairment of biosynthesis of PlsEtn and phosphatidylethanolamine [5]. In agreement with F741 fibroblasts analyzed in this study, the peroxisomal matrix protein import is normal in both cases. Therefore, dysmorphology of peroxisomes likely affects the PlsEtn biosynthesis despite normal peroxisomal localization of enzymes for the PlsEtn synthesis. Precise mechanism underlying the attenuation of PlsEtn biosynthesis in the cells with peroxisomal dysmorphology remains to be defined. A pathogenic defect, the decreased cerebellar foliation in the NME3 patient remains undefined [29]. We recently reported a similar developmental defect of cerebellum including abnormal foliation in a peroxisome biogenesis disorder mouse model, *Pex14**^Δ^**^C/**Δ**C^* mouse, manifesting nearly 50% reduction of the cerebellar PlsEtn level [54,55]. Therefore, the reduced level of PlsEtn is a potential cause of the defects in cerebellar development of the NME3 patient, although the PlsEtn level in the cerebellum of the patient is not determined.

## 4. Materials and Methods

### 4.1. Cell Culture, DNA Transfection, and RNAi

*PEX11β*^−/−^ MEF [12], HeLa cells and fibroblasts from a healthy control and an *NME3*-deficient patient were maintained in DMEM (Invitrogen, Carlsbad, CA, USA) supplemented with 10% FBS (Sigma, St. Louis, MO, USA). All cell lines were cultured at 37 °C under 5% CO_2_. DNA transfection was performed using Lipofectamine 2000 (Invitrogen) for HeLa cells according to the manufacturer’s instructions and cells were cultured for the indicated time periods.

si*RNA*-mediated knockdown of *NME3* and *ATAD1* in HeLa cells was performed using predesigned Stealth™ siRNAs (Invitrogen) and MISSION siRNA (Sigma), respectively. Cells were harvested or fixed at 72 h after the initial transfection using Lipofectamine 2000. The following siRNAs were used: human *NME3* #28 (NME3HSS143128), #29 (NME3HSS143129) and #30 (NME3HSS143130), and human *ATAD1* (SASI_Hs01_00200046).

### 4.2. Antibodies

Rabbit antibodies to acyl-CoA oxidase (AOx) [56], 3-ketoacyl-CoA thiolase [56], ADAPS [57], PTS1 peptide [58], Pex3 [45], Pex14 [59], Mff [11], DYNAMO1 [27], 70-kDa peroxisomal membrane protein (PMP70) [56], and guinea pig anti-Pex14 antibody [60] were as described. Rabbit polyclonal antibody, termed anti-NME3C antibody, was raised against the C-terminal 18-amino-acid sequence of human NME3. Rabbit antibody against NME3 was purchased from ABclonal (Tokyo, Japan). Mouse monoclonal antibodies against HA (16B12; Covance, Princeton, NJ, USA), β-actin (6D1; MBL, Nagoya, Japan), Tom20 (F-10; Santa Cruz Biotechnology, Dallas, TX, USA), DLP1 (BD Biosciences, Franklin Lake, NJ, USA), and LDHA (AF14A11; AbFrontier, Seoul, Korea) were purchased.

### 4.3. RT-PCR

Total RNA was isolated from HeLa cells using a TRIzol reagent (Ambion, Austin, TX, USA) and synthesis first-strand cDNA was performed with a PrimeScript RT reagent kit (Takara Bio, Kusatsu, Japan). Quantitative real-time RT-PCR was performed in an Mx3000P QPCR system (Agilent Technologies, Santa Clara, CA, USA) with SYBR Premix Ex Taq™ II (Ti RNaseH Plus) (Takara Bio). Primers used were: human *GAPDH* sense HsGAPDH_Fw. 5′-atggaaatcccatcaccatctt-3′, antisense HsGAPDH_Rv 5′-cgccccacttgattttgg-3′; human *NME3* sense NME_Fw. 5′-ctcatcggagccacgaacc-3′, antisense NME3_Rv. 5′-gttcttgccaacctcgatgc-3′; human *ATAD1* sense ATAD1_Fw. 5′-cctccaggctgtggtaaaac-3′, antisense ATAD1_Rv. 5′-agaagacagcagcagccaat-3′.

### 4.4. Plasmids

cDNA encoding full-length NME3 was amplified by PCR using the RT-product prepared from MCF7 cells as a template with a set of primers, HsNME3 Fw: 5′-GGGGATCCACCATGATCTGCCT GGTGCTGACCATC-3′ and HsNME3 Rv: 5′-CCCTCGAGCTCATACAGCCAGTGCCCAGCGCT GTC-3′, and cloned into *Bam*HI-*Xho*I sites of pcDNAZeo/C-HA_2_ [36]. To generate pcDNAZeo/*NME3*, DNA fragment encoding NME3 was amplified with a set of primers, T7 promoter Fw: 5′-TAATACGACTCACTATAGGG-3′ and HsNME3terRv: 5′-TTCTCGAGCTACTCATACAGC CAGTGCCCA-3′, and cloned into *Bam*HI-*Xho*I sites of pcDNAZeo. pcDNAZeo/*NME3H135A* was generated by inverse PCR using pcDNAZeo/NME3 as a template with a set of primers, HsNME3H135N Fw: 5′-AACGGCAGCGACTCGGTGG-3′ and HsNME3H135N Rv: 5′-TCAGGTTCTTGCCAACCTC-3′. The plasmids, pcDNAZeo/*HA-PEX26* and pcDNAZeo/ *FLAG-PEX19*, were as described [40].

### 4.5. Immunoblotting

Immunoblotting was performed as described [61]. In brief, protein samples were separated by SDS-PAGE and electrotransferred to a polyvinylidene fluoride membrane (Bio-Rad Laboratories, Hercules, CA, USA). After blocking in PBS containing 5% nonfat dry milk and 0.1% Tween 20, blots were subjected to immunoblotting with the indicated antibodies. Immunoblots were developed with ECL Western blotting detection reagents (GE Healthcare, Chicago, IL, USA), and scanned with an LAS-4000 Mini luminescent image analyzer (Fujifilm, Tokyo, Japan).

### 4.6. Immunofluorescence Microscopy

Cells on glass coverslips were fixed with 4% paraformaldehyde in PBS for 15 min at RT, permeabilized with 1% Triton X-100 in PBS for 2 min at RT, and blocked with PBS-BSA (PBS containing 1% BSA) for 30 min at RT. Subsequently, cells were incubated with primary antibodies indicated. Antigen-antibody complexes were visualized with Alexa Fluor conjugated secondary antibodies and observed as described [62]. Images were obtained under a laser-scanning confocal microscope (LSM710 with Axio Observer.Z1; Carl Zeiss, Oberkochen, Germany).

Number of peroxisomes per cell was determined in randomly selected cells. Peroxisomal number was calculated using the Particle Analysis package of Image J using a threshold images converted from optical images obtained by confocal fluorescence microscopy [7].

### 4.7. Subcellular Fractionation and Biochemical Analysis

Cells were harvested in buffer H (0.25 M sucrose, 20 mM Hepes-KOH, pH 7.4, 1 mM EDTA, and a protease inhibitor cocktail) and homogenized on ice by passing through a 27-gauge needle (with 1 mL syringe). Homogenates were centrifuged at 1000× *g* for 5 min to yield a post-nuclear supernatant (PNS) fraction. The PNS fraction was separated into cytosolic and organelle fractions by ultracentrifugation at 100,000× *g* for 30 min [63]. For subcellular fractionation, PNS fraction was centrifuged at 2500× *g* for 10 min to separate a heavy mitochondria fraction [11]. Post-HM fraction was further centrifuged at 100,000× *g* for 30 min to separate a light mitochondria and cytosol fractions.

Alkaline extraction was performed as described [64]. In brief, organelle fractions were treated with 0.1 M Na_2_CO_3_ on ice for 30 min, and separated into soluble and membrane fractions by ultracentrifugation at 100,000× *g* for 30 min.

In vitro transcription/translation reactions in a rabbit reticulocyte-lysate were performed using the TNT T7 Quick Coupled Transcription/Translation System (Promega, Madison, WI, USA) according to the manufacturer’s instruction.

### 4.8. Lipid Analysis

Analysis of PlsEtn in fibroblasts was performed as described [65] using 4000 Q-TRAP quadrupole linear ion trap hybrid mass spectrometer (AB Sciex, Foster City, CA, USA) with an ACQUITY UPLC System (Waters).

## Figures and Tables

**Figure 1 ijms-21-08040-f001:**
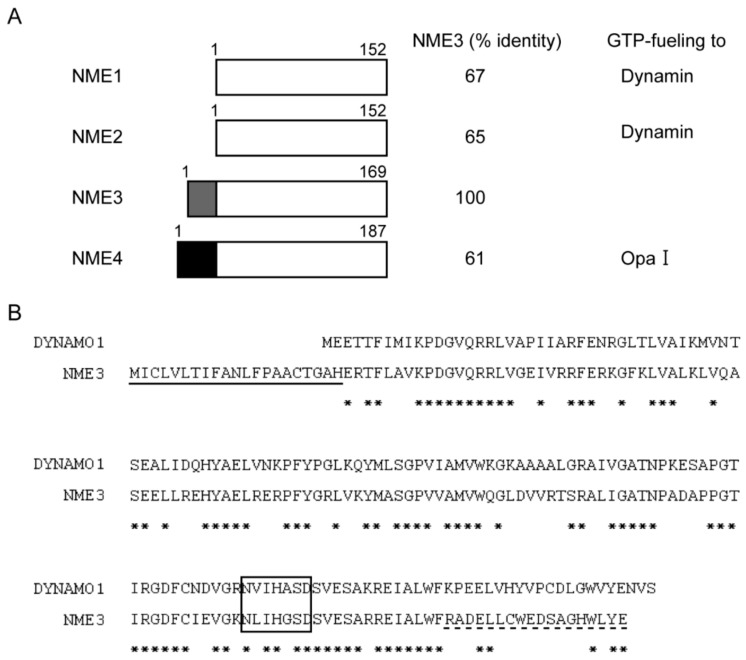
Mammalian nucleoside-diphosphate (NDP) kinase family. (**A**) A schematic view of four NDP kinase family members is from [39]. Amino-acid identity was also indicated between NME1, 2, 3, and 4 by taking NME3 as 100%. Proteins GTP-fueled by respective NME proteins are shown in the right column. Gray and solid boxes are N-terminal hydrophobic segment of NME3 and mitochondrial targeting signal of NME4, respectively. (**B**) Amino acid alignment of DYNAMO1 and NME3. *C. merolae* DYNAMO1 and human NME3 were aligned by using a Clustal W program. Identical amino acids are represented by asterisks; hydrophobic segment in the N-terminal region of NME3 is marked by an underline. The conserved NDP kinase active site is designated by a box. Amino-acid sequence used for generation of antibody is indicated by a broken-underline.

**Figure 2 ijms-21-08040-f002:**
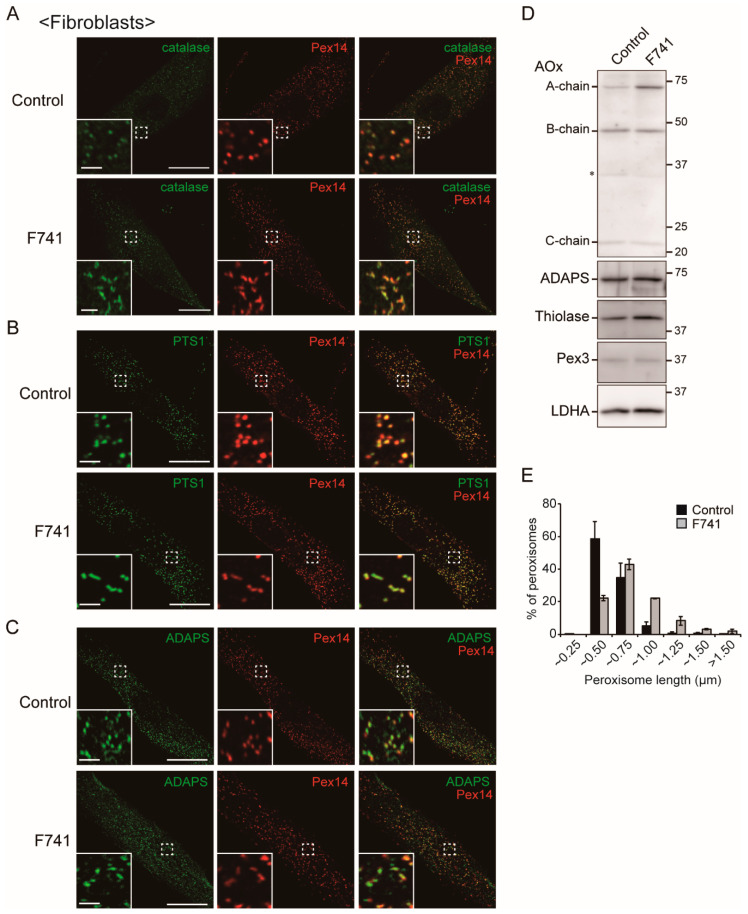
Aberrant morphology of peroxisomes in NME3-deficient fibroblasts. (**A**–**C**) peroxisomes in the fibroblasts from a healthy control (upper panels) and a patient F741 (lower panels) carrying a homozygous mutation in the initiation codon of *NME3* were visualized by indirect immunofluorescent staining with antibodies to catalase (**A**), PTS1 (**B**), ADAPS (**C**) and Pex14 (**A**–**C**). Insets show the images of the boxed areas. Bars, 20 µm and 2 µm (insets). Elongated peroxisomes are frequently observed in the patient-derived fibroblasts. (**D**) proteolytic processing of AOx, thiolase, and ADAPS was accessed by immunoblotting of cell-lysates of fibroblasts from a healthy control and the patient with antibodies to AOx, ADAPS, thiolase, Pex3, and LDHA, respectively. LDHA was used as a loading control. (**E**) histogram of peroxisome length measured in three each fibroblasts from a healthy control (1400 peroxisomes in three cells) and a patient F741 (1038 peroxisomes in three cells).

**Figure 3 ijms-21-08040-f003:**
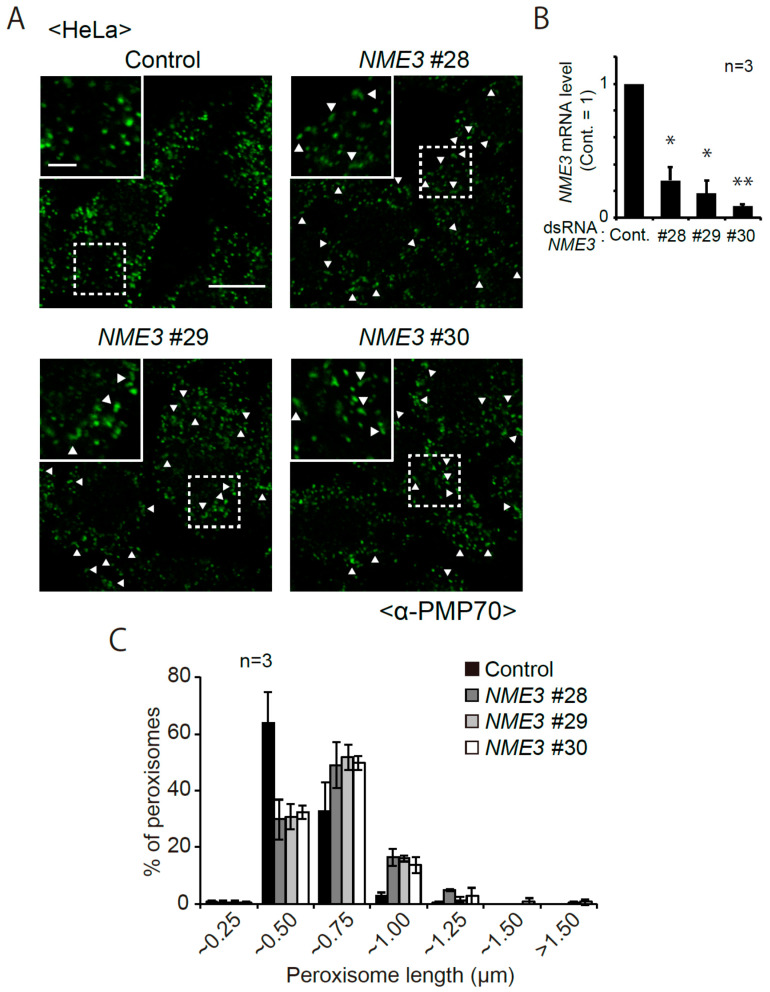
*NME3* knockdown elongates peroxisomes. (**A**) Three different dsRNAs (#28, #29, and #30) against *NME3* were transfected to HeLa cells as described in Materials and Methods. Morphology of peroxisomes was assessed with anti-PMP70 antibody. Scale bar, 10 µm. Higher magnification images of the boxed regions were shown (Inset). Scale bar, 3 µm. Arrowheads indicate elongated peroxisomes. (**B**) histogram of peroxisome length measured in three each control (totally 385 peroxisomes), ds*NME3* #28- (371 peroxisomes), ds*NME3*-#29-(349 peroxisomes), and ds*NME3*-#30-transfected (344 peroxisomes) HeLa cells. (**C**) mRNA level of *NME3* in HeLa cells that had been treated as in (**A**) was quantified by qRT-PCR (n = 3). Data indicate means ± SD. * *p* < 0.05, ** *p* < 0.01, by Student’s *t*-test.

**Figure 4 ijms-21-08040-f004:**
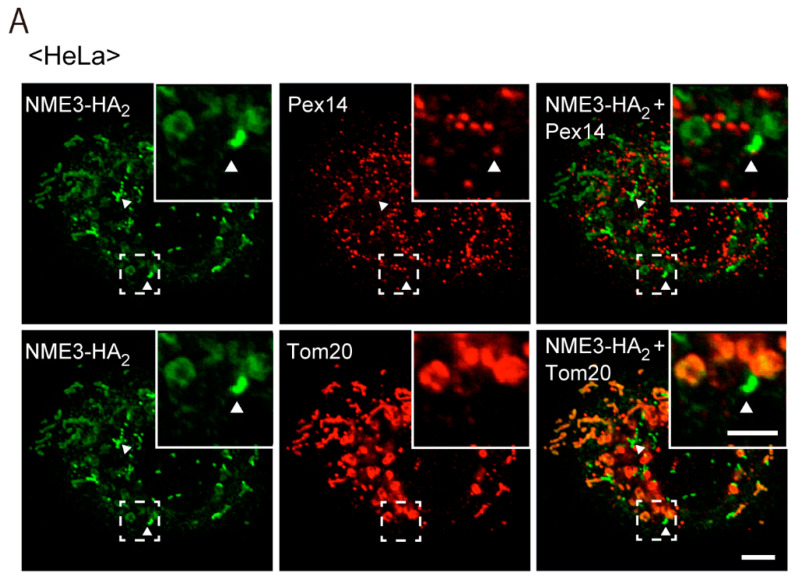
Intracellular localization of NME3-HA_2_ in HeLa cells. (**A**) NME3-HA_2_ was expressed in HeLa cells. NME3-HA_2_ was stained with rabbit anti-HA antibody (green). Peroxisomes and mitochondria were visualized with guinea pig anti-Pex14 (upper panels) and mouse anti-Tom20 (lower panels) antibodies, respectively. Peroxisomes are shown by a pseudo-color image. Scale bar, 10 µm. Higher magnification images of the boxed regions were shown (Inset). Scale bar, 5 µm. Arrowheads indicate peroxisomal, not mitochondrial, localization of NME3-HA_2_. (**B**) immunoblotting of mock- (-) and *NME3-HA_2_*- (+) transfected HeLa cells. Approximately ten times more of total proteins were loaded in lanes 1 and 4 than those in lanes 2 and 3. NME3-HA_2_ was detected with antibodies to HA (upper panel, lanes 1 and 2) and NME3 (ABclonal) (upper panel, lanes 3 and 4). Two bands (solid and open arrowheads) were detected and termed NME3-1-HA_2_ and NME3-2-HA_2_, respectively, by the expression of NME3-HA_2_. Dots indicate non-specific bands. β-actin, a loading control.

**Figure 5 ijms-21-08040-f005:**
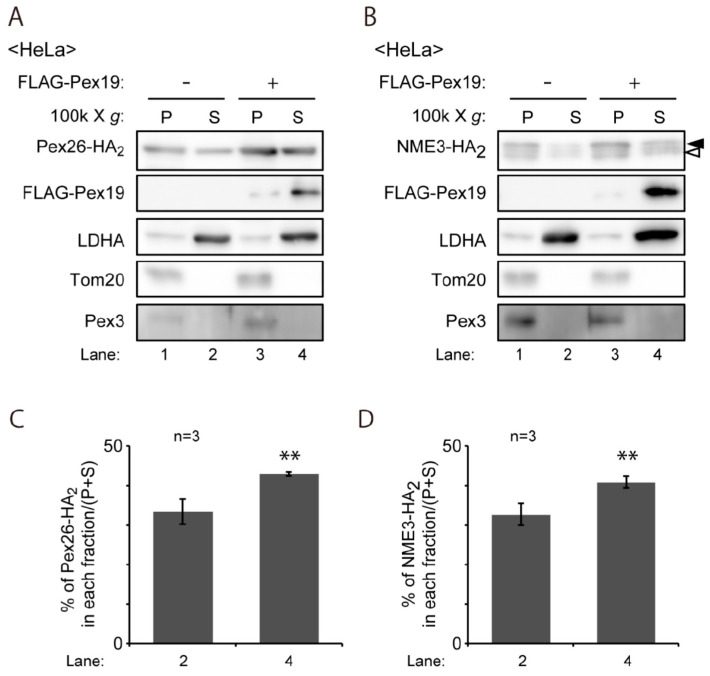
Overexpressed Pex19 stabilizes the coexpressed NME3-HA_2_ in the cytosol. (**A**) *PEX26-HA*_2_ was expressed alone or together with *FLAG-PEX19* for 20 h. Cells were fractionated into organelle pellet (P) and cytosol (S) fractions. Equal aliquots of each fraction were analyzed by SDS-PAGE and immunoblotting with antibodies to HA, FLAG, lactate dehydrogenase A (LDHA), a cytosolic protein; Tom20, a mitochondrial outer membrane protein; Pex3, a PMP. (**B**) NME3-HA_2_ was likewise expressed and assessed for its subcellular distribution as in (**A**). Two bands (solid and open arrowheads) were detected as in Figure 4B. (**C**,**D**), relative Pex26-HA_2_ and NME3-HA_2_ signals in cytosol fractions (S) were represented as a percentage of the total signal (S + P) (n = 3). ** *p* < 0.01, by Student’s *t*-test.

**Figure 6 ijms-21-08040-f006:**
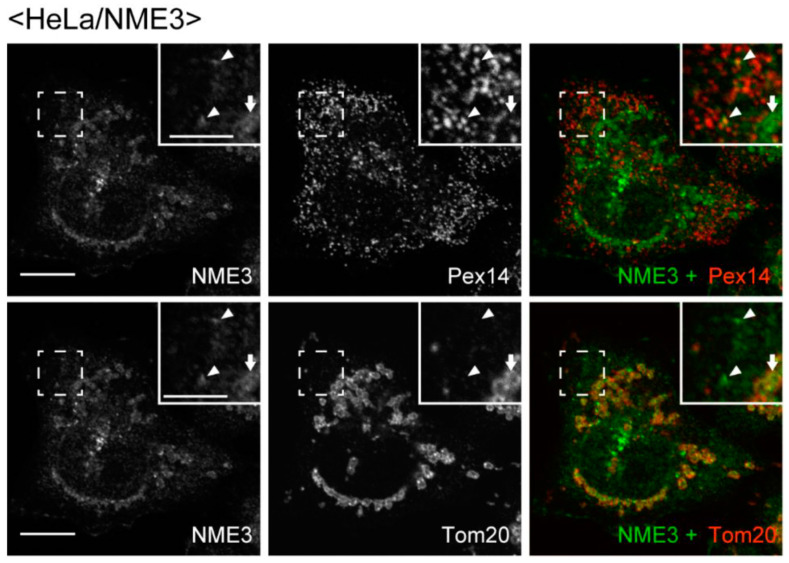
Intracellular localization of non-tagged NME3 in HeLa cells. Non-tagged NME3 was expressed in HeLa cells and stained with the antibody raised to DYNAMO1 (green). Peroxisomes and mitochondria were visualized with guinea pig anti-Pex14 (**upper panels**) and mouse anti-Tom20 (**lower panels**) antibodies, respectively. Peroxisomes are shown by a pseudo-color image. Scale bar, 10 µm. Higher magnification images of the boxed regions were shown (Inset). Scale bar, 5 µm. Arrowheads and arrows indicate peroxisomal and mitochondrial localization of NME3, respectively.

**Figure 7 ijms-21-08040-f007:**
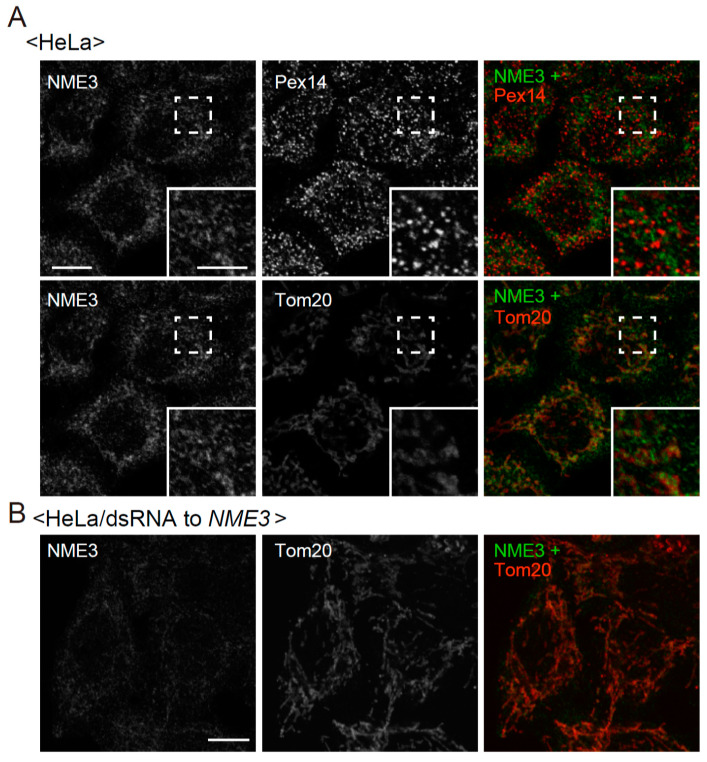
NME3 is localized to peroxisomes and involved in the fission of peroxisomes. (**A**) intracellular localization of endogenous NME3 in HeLa cells was assessed by indirect immunofluorescent cell staining with antibodies to DYNAMO1 (green), Pex14, and Tom20 as in Figure 6. Scale bar, 10 µm. Higher magnification images of the boxed regions are shown (Inset). Scale bar, 5 µm. Note that mitochondrial localization of NME3 is distinct (lower panels). (**B**) HeLa cells transfected with dsRNA against *NME3* (#28, see Figure 3) were verified by indirect immunofluorescent cell staining with anti-bodies to DYNAMO1 and Tom20. Scale bar, 10 µm. (**C**) HeLa cells transfected with dsRNA against *ATAD1* were verified by indirect immunofluorescent cell staining as in (**A**). Scale bar, 10 µm. Higher magnification images of the boxed regions are shown (Inset). Scale bar, 5 µm. Note that localization of NME3 in peroxisomes (arrowheads), but not mitochondria, is more readily discernible. (**D**) HeLa cells co-transfected with a set of two dsRNAs against *ATAD1* and *NME3* were verified by indirect immuno- fluorescent cell staining with antibodies to DYNAMO1 and Pex14. Higher magnification images of the boxed regions are shown (Inset). Note that peroxisomes are frequently elongated. Scale bar, 10 µm. Higher magnification images of the boxed regions were shown (Inset). Scale bar, 5 µm. (**E**) relative fluorescent intensity of NME3 in HeLa cells transfected with mock (-, n = 38) or dsRNA against *ATAD1* (+, n = 25) was quantified. * *p* < 0.05. (**F**,**G**) transcription level of *ATAD1* (**F**) and *NME3* (**G**) in HeLa cells treated as in (**C**) was quantified by quantitative real-time PCR (n = 3). (**H**) protein level of Pex14 was verified by immunoblotting (left). β-actin, a loading control. Protein level of Pex14 was represented as values relative to that in mock-treated HeLa cells (right, n = 3). (**I**) the number of peroxisomes in HeLa cells untreated (n = 37), transfected with dsRNA against *ATAD1* alone (n = 37), or a set of *ATAD1* and *NME3* (n = 33) was represented. *** *p* < 0.001, by Tukey-Kramer test. n.s., not significant. (**J**) NME3 level is elevated by ATAD1 knockdown. *siControl* and *siATAD1* were separately transfected twice with a 24 h interval to HeLa cells that had been transfected for 6 h with a plasmid encoding NME3. After 24-h cell culture, transcription level of *ATAD1* (left) was quantified by quantitative real-time PCR (n = 2). Center, NME3 expression levels were assessed by western blotting of respective cell lysates with anti-NME3C antibody. Two bands, NME3-1 and NME3-2, marked by solid and open arrowheads were detected. β-actin, a loading control. Dots indicate non-specific bands; right, NME3-1 and NME3-2 bands were quantified and represented by taking as 1 NME3-1 in *siControl*-transfected cells in lane 1. (**K**) upper row, NME3 was expressed in *PEX11β*^−/−^ mouse embryonic fibroblasts (MEF) [12] and its intracellular localization was verified by staining with antibodies to DYNAMO1 (**a**), DLP1 (**b**), and Pex14 (**c**). Scale bar, 10 µm. Higher magnification images of the boxed regions were shown (Inset). Scale bar, 5 µm. Note that NME3 was detected in the limited area of an elongated peroxisome (arrowhead). Panel (**e**), signal intensity of NME3, DLP1, and Pex14 in the elongated peroxisome indicated with its length in the merged view was analyzed by line scanning and represented. Note that NME3 was localized at the DLP1-accumulated potential constriction site (arrowhead in (**a**–**d**)) where signal of Pex14 is weak and the both sides adjacent to this region the constriction site.

**Figure 8 ijms-21-08040-f008:**
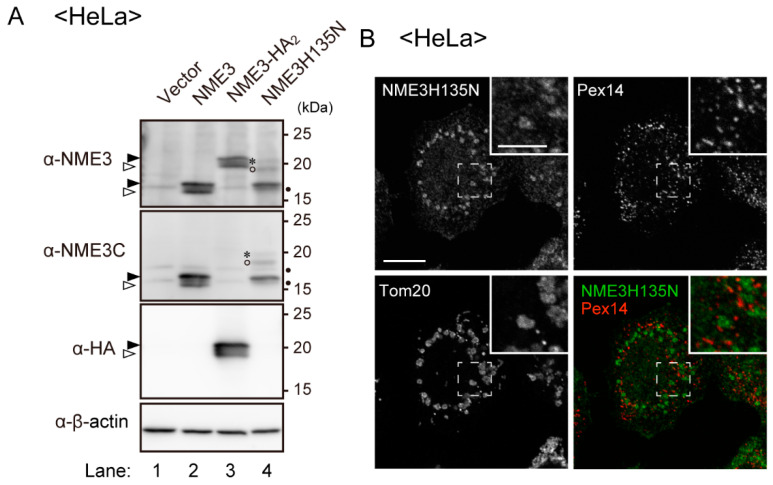
NDP kinase activity of NME3 is essential for the division of peroxisomes. (**A**) ectopic expression of NME3 and its mutant NME3H135N in HeLa cells was verified by immunoblotting of respective cell lysates with antibodies to NME3 and NME3C. NME3-HA_2_ was likewise expressed and assessed with anti-HA antibody. Two bands, NME3-1, NME3-2, and HA_2_-tagged ones, respectively, marked by solid and open arrowheads were detected (lanes 2 and 3); three bands, NME3H135N, NME3H135N-a, and NME3H135N-b were marked by solid arrowhead, asterisk, and open circle, respectively. β-actin, a loading control. Dots indicate non-specific bands. (**B**) peroxisome morphology was verified by expressing NME3H135N in HeLa cells. Cell staining was performed as in Figure 6. Scale bar, 10 µm. Higher magnification image of the boxed regions was shown (Inset). Scale bar, 5 µm. (**C**) distribution of NME3 in heavy mitochondrial (HM) and post-HM membrane fractions containing light mitochondria (LM) and microsome (Ms) fractions. PNS fractions prepared from HeLa cells (lane 2 in (**A**)) were fractionated as described in Materials and Methods and assessed for distribution of NME3 and marker proteins for mitochondria and peroxisomes with antibodies indicated on the left. Two bands indicated by solid and open arrowheads are as in (**A**). Dots, non-specific bands. (**D**) subcellular distribution of NME3 and marker proteins was represented in each fraction by taking as 1 total amount of respective proteins detected in lanes 2–4 (*n* = 3). Solid, gray, and open bars indicate the levels in HM, LM plus Ms, and cytosol fractions, respectively. The values were represented as means ± SD of three independent experiments (upper panel) and a single experiment (lower panel), respectively. NME3 was mostly recovered in both HM (solid bar) and post-HM fractions (gray bar), where NME3-2 (open arrowhead in (**C**)) was mainly recovered in the post-HM fraction. Note that NME3 was detected in a manner of a dual distribution in mitochondria and peroxisome fractions, similar to dually localized Mff. (**E**) organelle fraction (100,000× *g* pellet fraction of PNS) from HeLa cells expressing a higher level of NME3 was treated with 0.1 M Na_2_CO_3_ and separated into soluble (S) and membrane (M) fractions. Equal aliquots of respective fractions were analyzed by immunoblotting with the indicated antibodies. Solid and open arrowheads are as in (**C**). (**F**) cell lysates prepared from HeLa cells expressing the NME3 variants and cell-free translated NME3 and the mutant NME3H135N were analyzed in the same SDS-PAGE gel and detected with NME3C antibody. Solid and open arrowheads, asterisk, open circle, and a dot are as in (**A**).

**Figure 9 ijms-21-08040-f009:**
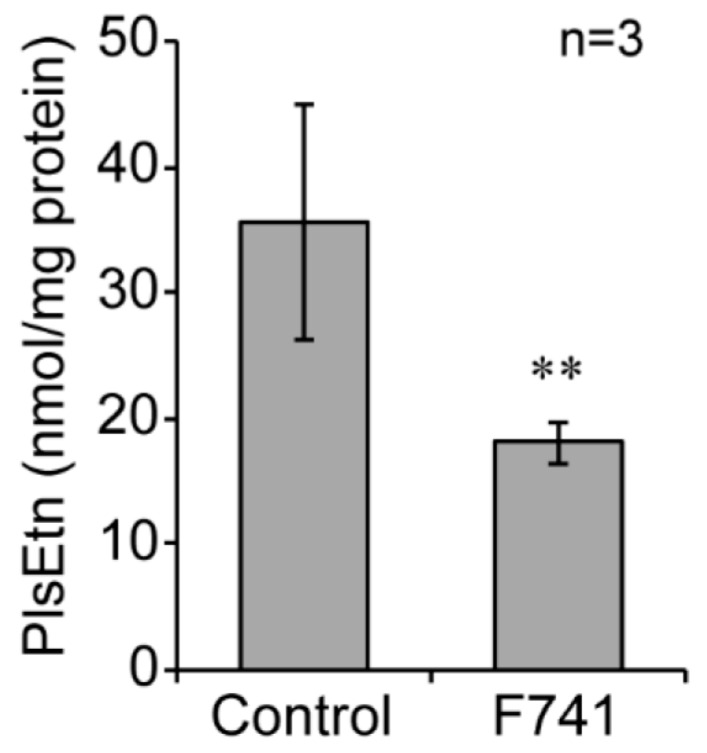
Plasmalogens are decreased in fibroblasts from a NME3-deficient patient. Ethanolamine plasmalogens (PlsEtn) in fibroblasts from a healthy control and the patient F741 were determined by LC-MS/MS (n = 3). Data indicate means ± SD. ** *p* < 0.01, by Student’s *t*-test.

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
