# Peer review of "Mammalian Homologue NME3 of DYNAMO1 Regulates Peroxisome Division"

_ijms, 2020, doi:10.3390/ijms21218040_

Round 1
Reviewer 1 Report
Dear Masanori Honsho and colleagues,
I read your manuscript on the potentially novel role of the nucleoside diphosphate kinase NME3 in peroxisome division. While I agree with the authors that there is a potentially new role for this protein in the life cycle of the peroxisome, I feel that the evidence is not yet strong enough to make this claim.
Here are my comments how the experimental evidence could be improved:
Figure 1: since the F741 cell line has a mutation in the initiation codon of the nme3 gene and aberrant morphology of the peroxisomes - it would be good to show examples of aberrant peroxisomes at a higher magnification as the small inserts in Fig 1 are difficult to see - which should correlate with the absence of the NME3 protein, it is important to show a Western blot of the control and F741 cells probed with the anti-NME3 and anti-NME3-C antibodies. Having a mutation in the initiation codon may render a downstream AUG codon in an alternative initiation codon resulting in a N-terminally truncated NME3 protein.
Figure 3: as above, please include a Western blot
Figure 4: while the colocalisation of NME3-HA2 colocalises well with Tom20. the colocalisation of NME3-HA2 with Pex14 is only limited to a few spots or in other words to a few peroxisomes. As this might indicate that NME3 localises to dividing peroxosomes, it would be important to counterstain the NME3-labelled organelles with a peroxisome division protein like MFF or Pex11beta or DLP1.
Figure 7: a Westernblot of HeLa, HeLa dsRNA-NME3 and HeLa dsRNA-ATAD1 is missing
Figure 8A: I suggest to number the different bands of NME3 as it is confusing when the slower or faster migrating bands are discussed
Figure 8B: a panel of cells transected with WT NME3 is missing - it is interesting to see that this partial colocalisation of NME3 with Pex14 is gone in the catalytic inactive mutant H135N
Figure 8C: a panel probed with the normal anti-NME3 antibody is missing - it would also be interesting to repeat this experiment with the WT and H135N mutated NME3 in the F741 cell line
Figure 8D: please include color legend in the image panel
Figure 8E (is missing from the figure legend) I find that this a very important assay - please probe the extracts with the anti-TOM20 antibody and repeat the test in the dsRNA-ATAD1 and NME3-H135N OE cell lines
Figure 9: please repeat with OE WT NME3 and the NME3-H135N cells in control and F741 backgrounds
Author Response
We thank you for taking time to review the manuscript and suggesting the critical and helpful comments.
Replies to comment 1:
Figure 2? 1: since the F741 cell line has a mutation in the initiation codon of the nme3 gene and aberrant morphology of the peroxisomes - it would be good to show examples of aberrant peroxisomes at a higher magnification as the small inserts in Fig 1 are difficult to see - which should correlate with the absence of the NME3 protein, it is important to show a Western blot of the control and F741 cells probed with the anti-NME3 and anti-NME3-C antibodies. Having a mutation in the initiation codon may render a downstream AUG codon in an alternative initiation codon resulting in a N-terminally truncated NME3 protein.
In Figure 2A-C, we replaced the higher magnification images as suggested.
We attempted to detect the endogenous NME3 protein by western blotting using anti-NME3 and anti-NME3C. However, the antibodies used did not recognize the endogenous NME3 in fibroblasts from a healthy control and HeLa cells. Our co-authors reported that the shorter NME3 protein derived from the F741 patient presumably translated from the second AUG codon at the position of Met85 in the open reading frame was not detected by introducing GTG mutation at ATG of Met1 in the plasmid encoding NME3 (a cited reference #29: Chen, CW et al. 2019 PNAS 116: 566-574). Together, these results indicate that the expression level of a shorter form, if any, of NME3 in F741 is very low.
Replies to comment 2:
Figure 3: as above, please include a Western blot
We attempted to analyze the NME3 protein level by western blotting. However, the antibodies used in this study apparently failed to cross-react with endogenous NME3 in HeLa cells, as we described above. Instead, we showed the reduction of NME3 mRNA expression by transfecting three different dsRNAs against NME3, where peroxisomes were elongated as observed in the NME3-lacking F741 cells. Thus, we believe that our data adequately provided the results regarding the effect of the reduced expression of NME3 on the morphology of peroxisomes.
Replies to comment 3:
Figure 4: while the colocalisation of NME3-HA2 colocalises well with Tom20. the colocalisation of NME3-HA2 with Pex14 is only limited to a few spots or in other words to a few peroxisomes. As this might indicate that NME3 localises to dividing peroxosomes, it would be important to counterstain the NME3-labelled organelles with a peroxisome division protein like MFF or Pex11beta or DLP1.
We thank you for the helpful comment. We investigated the localization of NME3 in PEX11b-knocked out (PEX11β-/-) MEF cells where peroxisomes are elongated due to the absence of Pex11b (reference #12). We found that NME3 was accumulated in an elongated peroxisome localized with DLP1 in the PEX11β-/- MEF cells. We added these results in Figure 7K and described on page 10, lines 226-231.
Replies to comment 4:
Figure 7: a Westernblot of HeLa, HeLa dsRNA-NME3 and HeLa dsRNA-ATAD1 is missing
In the submitted manuscript, we presented the data representing the protein level of endogenous NME3 in HeLa and HeLa cells transfected with ATAD1 dsRNA by quantifying the immunofluorescence signal of NME3 (Figure 7D; 7E in the revised Ms) together with the transcription level of ATAD1 and NME3 (Figure 7E, F; 7F, G in the revised Ms). With the anti-NME3 antibodies used, we were unable to detect the NME3 protein by western blotting.
Just in case, we are afraid of your reviewing the submitted manuscript that was formatted by Journal editorial office, where the initially submitted Figures 7D to 7H were missing at position of page 11, line 221. Anyhow, there were no western blotting performed for NME3.
In the revised manuscript, the elevation of NME3 by ATAD1 knockdown was confirmed by ectopic expression of NME3 in HeLa cells, where about 60% decrease in ATAD1 mRNA and ~1.4-fold increase in NME3-1 were detectable (Fig. 7J), consistent with the profile of endogenous NME3 (Fig. 7E-G, I).
Reply to comment 5:
Figure 8A: I suggest to number the different bands of NME3 as it is confusing when the slower or faster migrating bands are discussed
The lower and higher mobility bands of NME3 were termed as NME3-1 and NME3-2, respectively, throughout the manuscript.
Replies to comment 6:
Figure 8B: a panel of cells transected with WT NME3 is missing - it is interesting to see that this partial colocalisation of NME3 with Pex14 is gone in the catalytic inactive mutant H135N
In Figure 6 of the initially submitted manuscript, we showed mitochondrial and peroxisomal localization of ectopically expressed NME3 in HeLa cells.
We provided new data showing the subcellular distribution of NME3H135N in Supplementary Figure S1 where NME3H135N was mostly recovered in HM-fraction similar to NME3-1 and Tom20 shown in Figure 8, C and D. These results together with the immunofluorescence data of NME3H135N indicate that NME3H135N is mostly localized to mitochondria and partly to post-HM fractions.
Replies to comment 7:
Figure 8C: a panel probed with the normal anti-NME3 antibody is missing - it would also be interesting to repeat this experiment with the WT and H135N mutated NME3 in the F741 cell line
As shown in Figure 8A, the western blot signal of NME3 detected with the normal anti-NME3 is basically the same as that probed with anti-NME3C. Therefore, we think that the results analyzing the subcellular distribution of NME3 probed with the normal anti-NME3 are unnecessary.
Analysis of subcellular distribution of NME3 and NME3H135N in F741 fibroblasts might be of interest. However, F741 fibroblasts were not readily accessible to our electroporation condition for introducing plasmid DNA. Furthermore, transfection of plasmid DNA to the fibroblasts by electroporation or liposome-mediated method was insufficient to verify the subcellular distribution of NME3 proteins by biochemical method. Instead, we might be able to use NME3-knock out HeLa cells. However, it will take longer than two months to isolate and characterize the NME3-defective cell line. With these technical limitations, we were unable to address the subcellular distribution of NME3 and NME3H135N in the F741 fibroblasts.
We rather showed that NME3 was localized to both mitochondria and peroxisomes in ATAD1-knockdown cells (Figure 7C). The same intracellular localization of ectopically expressed NME3 was obtained in HeLa cells (Figure 8C, D). These results suggest that NME3 is localized to both mitochondria and peroxisomes even in cells expressing a higher level of NME3. Similarly, NME3H135N was mostly localized to mitochondria as shown in Supplementary Figure S1. Therefore, we think that intracellular localization of NME3 and NME3H135N won’t be altered in F741 fibroblasts.
Reply to comment 8:
Figure 8D: please include color legend in the image panel
We added color legend in the panel of Figure 8D.
Replies to comment 9:
Figure 8E (is missing from the figure legend) I find that this a very important assay - please probe the extracts with the anti-TOM20 antibody and repeat the test in the dsRNA-ATAD1 and NME3-H135N OE cell lines
We added the western blot data probed with Tom20 antibody in Figure 8E.
We also provided the new data regarding the sodium carbonate extraction results of NMEH135N prepared from membrane fractions derived from Hela cells knocking down of ATAD1 and those expressing NME3-H135N in Supplementary Figure S1.
Replies to comment 10:
Figure 9: please repeat with OE WT NME3 and the NME3-H135N cells in control and F741 backgrounds
This suggestion is of interest. However, we were unable to assess the level of PlsEtn upon expressing NME3 or NME3H135N in control and F741 fibroblasts, due to the low transfection efficiency as described in Replies to comment 7.
Reviewer 2 Report
This is a well-designed study with the appropriate controls.
I have only some suggestions:
-1 Is there some possibility to quantify the number of elongated peroxisomes induced by NME downregulation? It is indeed quite important to have a quantitative and not only a qualitative evaluation of the role of NME3 on peroxisome structure.
-2 The authors found that in the absence of NME3 the peroxisome elongation is accompanied by a decrease of about 50% in the total amount of plasmalogens.
Have this observation any relevance in the pathology of the people carrying the homozygous mutation of NME3? Please discuss.
Author Response
We thank you for taking time to review the manuscript and suggesting the helpful comments.
Replies to comment 1:
-1 Is there some possibility to quantify the number of elongated peroxisomes induced by NME downregulation? It is indeed quite important to have a quantitative and not only a qualitative evaluation of the role of NME3 on peroxisome structure.
We measured the peroxisome length in F741 patient’s fibroblasts and NME3-knocked down HeLa cells and showed the histograms of the distribution of elongated peroxisomes with respective sizes in Figures 2E and 3C, respectively.
Replies to comment 2:
-2 The authors found that in the absence of NME3 the peroxisome elongation is accompanied by a decrease of about 50% in the total amount of plasmalogens.
Have this observation any relevance in the pathology of the people carrying the homozygous mutation of NME3? Please discuss.
We added new sentences in the last paragraph of Discussion about a potential cause of the defects in cerebellar pathology of the NME3-deficient patient likely resulted by the reduced ethanolamine plasmalogen level.
Round 2
Reviewer 1 Report
Dear authors,
thank you very much for your thorough response to my comments. Your changes clearly improved the paper. I wish you good luck for the future work.